# Correction of PM$_{2.5}$ Underestimation in Low-Cost Sensors under Elevated Dust Loading Using Only Sensor Measurements

Kamaljeet Kaur[1], Tristalee Mangin[1], Kerry E. Kelly[1]

[1]Department of Chemical Engineering, University of Utah, Salt Lake City, UT, 84112, USA

*Correspondence to*: Kamaljeet Kaur (kamaljeet.kaur@chemeng.utah.edu)

**Abstract.** The Plantower PMS5003/6003 sensor is widely used for low-cost monitoring of particulate matter (PM), but it substantially underestimates PM$_{2.5}$ and PM$_{10}$ during periods of elevated dust loading, when the particle size distribution is dominated by particles > 1 μm in diameter. This limitation is especially critical in the arid regions, such as the western United States, where windblown dust frequently degrades air quality, visibility, and public health. Accurate estimation of PM$_{2.5}$ and PM$_{10}$ concentrations during periods dominated by dust typically relies on federal reference or equivalent methods (FRM/FEM), but these resources have limited spatial resolution. This study investigates whether PMS5003/6003 measurements alone can be used to detect and to bias correct for these dust-dominant PM conditions. We analyzed measurements from 109 PMS sensors collocated or near 75 U.S. EPA monitoring sites with hourly FEM PM$_{2.5}$ and/or PM$_{10}$ between January 2017 to May 2025. Two cutoff thresholds (threshold1 and threshold2) were developed using relative humidity and the sensor-reported ratio of coarse (2.5–10 μm) to submicron (0.3–1 μm) mass concentration to identify potential periods dominated by dust when the PMS sensor underestimated PM$_{2.5}$ concentration. The thresholds can be used in real time, relying on the preceding 336 hourly measurements (consistent with PurpleAir's public archive display). To improve PM$_{2.5}$ estimates from the PMS sensor (pm2.5_alt, a common correction for Plantower PMS measurements reported by PurpleAir), this study used pm2.5_alt measurements identified as potential dust-dominated periods to develop a correction factor through non-linear regression. This correction reduced the mean bias error between PMS PM$_{2.5}$ estimates (pm2.5_alt) and FEM PM$_{2.5}$ by approximately 50% for 97 sensors, and reduced the root mean square error by approximately 30% for 84 sensors. This framework enhances the utility of PMS5003/6003 measurements during periods of elevated dust loading, extending monitoring capabilities in regions where regulatory coverage is limited.

## 1 Introduction

Plantower particulate matter sensors (PMS) are among the most widely used low-cost sensors for measuring particulate matter (PM) in ambient and indoor air (Barkjohn et al., 2021; Jaffe et al., 2023; Kim et al., 2025; Searle et al., 2023; Wallace et al., 2022). The PurpleAir (PA) network uses PMS sensors, and as of August 12, 2025, it had deployed 26,055 nodes worldwide (Worldwide, Ranked Data by PurpleAir, 2025). Additionally, other sensor networks use PMS sensors (Air Quality Egg - Science is Collaboration, 2025; Outdoor Air Quality Monitor, 2025; Air Quality Monitoring. Monitor in UK & Europe. Airly Data Platform and Monitors | Airly, 2025; Low-Cost Air Quality Monitoring & Measurement | Clarity Movement Co., 2025; QuantAQ, 2025; TELLUS Air Quality Monitoring Data and Solutions, 2025). The PMS5003 found in the PA-II is one of the most commonly used PM$_{2.5}$ sensors; this sensor reports particle number concentrations in six size bins, along with PM$_1$ (PM with aerodynamic diameter less than 1 μm), PM$_{2.5}$ (PM with aerodynamic diameter less than 2.5 μm), PM$_{10}$ (PM with aerodynamic diameter less than 10 μm), temperature (T), and relative humidity (RH) measurements (Barkjohn et al., 2021; Kaur and Kelly, 2023a; Sayahi et al., 2019; Searle et al., 2023). The PA-II typically contains two sensors per node, which provides an indication of measurement consistency (Barkjohn et al., 2021).

The PMS5003 sensor has been extensively studied, and numerous correction factors have been developed to improve its $PM_{2.5}$ measurement accuracy under varying environmental conditions (Ardon-Dryer et al., 2020; Barkjohn et al., 2021, 2022; Cowell et al., 2022; Hong et al., 2021; Hua et al., 2021; Jaffe et al., 2023; Kaur and Kelly, 2023b; Magi et al., 2020; Mai et al., 2025; Malings et al., 2020; Mathieu-Campbell et al., 2024; Nilson et al., 2022; Patel et al., 2024; Raheja et al., 2023; Si et al., 2020; Tryner et al., 2020b; Wallace, 2023; Wallace et al., 2022; Weissert et al., 2025). Some of the most commonly used correction algorithms, such as pm2.5_alt and Barkjohn's U.S. universal correction, can be integrated into PurpleAir's real-time maps (Barkjohn et al., 2021; Wallace, 2023). However, a persistent limitation of the PMS sensors is their inability to accurately detect larger particles (roughly >1 μm in diameter) (Gautam et al., 2025a; He et al., 2020; Kaur & Kelly, 2023a; Kosmopoulos et al., 2023; Kuula et al., 2020, Ouimette et al., 2022; Tryner, Mehaffy, et al., 2020), which leads to a significant underestimation of sensor-reported $PM_{2.5}$ and $PM_{10}$ concentrations during periods dominated by large particles, such as dust (Gautam et al., 2025b; Jaffe et al., 2023; Kaur and Kelly, 2023b; Masic et al., 2020; Vogt et al., 2021; Weissert et al., 2025). This significant underestimation of PM levels during dust-dominant periods can be misleading. For example, looking at a map of $PM_{2.5}$ concentrations from a source that utilizes Plantower PMS sensors during a dust-dominant PM episode will likely show inaccurately low PM concentrations, which can lead individuals and decision makers to significantly underestimate the associated health risks. Over the long term, this can lead to distrust in these low-cost sensor networks. Prior studies (Kuula et al., 2020; Ouimette et al., 2024; Ouimette et al., 2022) have demonstrated that this underestimation of coarse PM is due to scattering truncation error. This occurs because the photodiode in the sensor is poorly positioned to detect forward-scattered light, which is dominant for large particles, meaning the sensor often fails to register them effectively.

Identifying periods dominated by dust typically relies on FEM $PM_{2.5}$ and $PM_{10}$ measurements to obtain an estimate of coarse PM, which can be supplemented by satellite imagery, visibility reports, and high wind speed indicators (Hand et al., 2017; Jaffe et al., 2023; Kaur and Kelly, 2023b; Robinson and Ardon-Dryer, 2024; Sandhu et al., 2024; Tong et al., 2012). However, the availability of FEM, FRM, and other high-quality measurements is limited. For example, the continental United States has 2,141 $PM_{2.5}$ and 672 $PM_{10}$ monitoring sites, respectively, with only 502 locations operating both $PM_{2.5}$ and $PM_{10}$ monitors (AirNow-Tech: Home, 2025). In addition, satellite products are typically available after a time delay; for example, MODIS data is usually available 60 to 125 minutes after the satellite observation (MODIS Near Real-Time Data | NASA Earthdata, 2025). Thus, relying on FEM/FRM measurements or satellite products is impractical for large-scale or real-time applications (Brahney et al., 2024). To address these limitations, some studies have used nearby FEM sites to estimate correction factors for PMS sensors (Weissert et al., 2025). These methods often rely on rolling 3- or 4-day correlations or FEM-to-PMS concentration ratios, but such strategies have limitations, including limited spatial representativeness and time lags. Dust-dominated PM conditions typically last a few hours, as they are associated with short-term elevated wind speeds and larger particles settle quickly (Brahney et al., 2024). Common strategies for correcting low-cost PM sensor measurements, such as a rolling 3- or 4-day average, cannot be implemented in real time and may fail to capture periods dominated by larger particles because the averaging window can include both dust-dominated periods and periods dominated by other sources.

Recognizing periods dominated by dust is important in western regions of the United States, where such episodes can occur frequently due to arid landscapes, land disturbance, sparse vegetation, and high wind activity. These regions, including parts of California, Arizona, Utah, New Mexico, and Texas, experience elevated levels of windblown dust, which can significantly impact air quality, visibility, and public health (Ardon-Dryer et al., 2023a, b; Goudie, 2014; Hahnenberger and Nicoll, 2012; Kaur et al., 2025; Lei et al., 2016; Lewis et al., 2011; Robinson and Ardon-Dryer, 2024). Accurate detection and correction of dust-related pollution are therefore essential for both regulatory monitoring and public exposure assessments (Ardon-Dryer et al., 2023a).

This manuscript examines whether PMS measurements alone can be used to identify and bias-correct their measurements during periods dominated by larger particles (e.g., dust), without reliance on external data such as FEM monitors, satellite imagery, or meteorological information. This question is inspired by findings from Ouimette et al.

(2024), who noted that although the PMS sensor is often described as a nephelometer, it actually counts individual
particles. In the PMS sensor, the probability of detection increases with particle size, meaning larger particles are more
likely to be counted. However, the PMS does not effectively size the large particles (>1 µm) to the correct bin, due to
scattering truncation errors. Correctly sizing these larger particles depends on their passage through the sensor's focal
point, where sufficient light scattering occurs to be detected by the photodiode. Ouimette et al. (2024) estimated that
the probability of a 10 µm particle being correctly sized in a PMS sensor to produce a detectable signal is less than
2%. Consequently, on days with relatively low PM levels, counts in the coarse particle bin (2.5–10 µm) are expected
to be negligible due to both the scarcity of coarse particles and the low probability of correct bin assignment. Even
with the low probability of correct classification, during dust-dominated periods, the coarse bin registers higher counts
compared to clean days, reflecting the increased presence of coarse particles and providing a potential pathway for
identifying periods dominated by dust using the sensor alone. Building on this rationale, Jaffe et al. (2023) proposed
using the ratio of 0.3 µm to 5 µm PMS bin counts as a dust indicator, suggesting a cutoff value of 190, below which
measurements were likely associated with dust events, and suggested a correction method using the measurements
from one site, Keeler, California. Their method improved corrected PMS $PM_{2.5}$ measurements during dust events at
this single controlled site (operated by the air quality agency), but it did not provide a useful correction for most of
the 50 other sensors, collocated at monitoring stations (Jaffe et al., 2023).
Building on these previous studies (Jaffe et al., 2023; Ouimette et al., 2024), this study developed sensor-specific
parameters for identifying potential periods dominated by dust, when PMS sensors severely underestimate $PM_{2.5}$,
derived solely from PMS sensor measurements, without relying on external data sources. By analyzing internal
metrics, such as particle count distributions and RH, it provides a framework that can be applied to any PMS5003
sensor, regardless of location, to identify potential periods dominated by dust and to bias correct the sensor
measurements of $PM_{2.5}$ concentration. This approach expands the usability of the vast network of publicly available
sensors during periods dominated by dust, even in areas where regulatory monitoring is lacking.

## 2 Method

This section describes the PMS sensor, the data sources, and the time periods used in this study. It also describes the
PMS sensor data cleaning procedures, the sensor parameters of interest, the post-processing and real-time approaches
for identifying potential periods dominated by dust and for bias correcting the sensors' underestimates of $PM_{2.5}$, as
well as the statistical tools used for data analysis. This study focuses on identifying conditions that are specifically
associated with PMS underestimation. It does not attempt to identify "dust events" in part because there is no well-
defined dust event classification method based solely on FEM $PM_{10}$ and $PM_{2.5}$ measurements. Moreover, the PMS
sensor's performance depends strongly on the underlying particle size distribution (Kaur and Kelly, 2023a; Kuula et
al., 2020; Ouimette et al., 2024). Although the PMS sensors are inefficient at measuring particles with diameters > 1
µm (Kaur and Kelly, 2023a; Kuula et al., 2020; Ouimette et al., 2024), the PMS sensor can still provide reasonable
estimates of $PM_{2.5}$ concentrations when concentrations of particles > 1 µm in diameter are elevated, as well as particles
≤ 1 µm in diameter.

### 2.1 Plantower PMS5003 and PMS6003 sensors

Several studies have described the Plantower PMS5003 sensors and their laboratory and field performance (Barkjohn
et al., 2021, 2022; Ouimette et al., 2024; Ouimette et al., 2022; Sayahi et al., 2019). PMS5003 uses a fan to create
flow (~1.67 mL/sec), a red laser (~680 +/- 10 nm), a scattering angle of 90°, and a photo-diode detector to measure
total scattering from a plume of particles (Kaur & Kelly, 2023a; Ouimette et al., 2022). The sensor converts the total
light scattering into several different air quality parameters, including particle counts in six bins (>0.3 µm, >0.5 µm,
>1 µm, >2.5 µm, >5 µm, and >10 µm), and $PM_1$, $PM_{2.5}$, and $PM_{10}$ using an embedded algorithm. The flow path
involves more than one 90° turn before particles reach the photodiode. Several other models of the Plantower PMS
sensor exist (i.e., PMS1003, 3003, 6003, 7003, 9003, A003, T003, X003). Kaur and Kelly (2023a) evaluated PMS6003
and found that the PMS5003 and PMS6003 exhibited similar performance to coarse PM. Many of the PMS models
have similar configurations and likely exhibit similar challenges with accurately measuring coarse PM, although this
has not been systematically evaluated.
This study used the PurpleAir network PMS sensors, i.e., PA-II. This study period began in 2017 and spanned several
years, during which time the PA-II nodes came in different configurations (PA-II, PA-II-SD, and PA-II-FLEX),
employed two different Plantower PMS sensors (PA-II and PA-II-SD: 5003 and PA-II-FLEX: 6003), and used different
firmware versions (6.06b, 7.02, and 7.04). Due to the lack of detailed documentation on how different firmware
versions affected sensor performance, no firmware-based exclusions were made. The PA-II-SD model is a PA-II sensor
variant that includes an SD card for data storage; both of these variations were included in the study. The PMS6003,
used in PA-II-FLEX, differs from PMS5003, primarily in the number of lasers used (as described in Kaur and Kelly
(2023a)), but its flow design, performance, and overall configuration are similar to the PMS5003 (Kaur and Kelly,
2023a). Accordingly, PA-II-FLEX data were not treated differently in this analysis. The ratio of >0.5 μm to 0.3 μm
(ratio greater than 0.4) was used to identify and exclude these alternate PMS5003, i.e., a PMS5003 version appeared
in June 2021 for a limited period of time and exhibited $PM_{2.5}$ concentrations that were biased low (Searle et al., 2023).
For the remaining part of the manuscript, the sensors will be referred to as PMS sensors.

**2.2 Sensor selection, data access, and cleaning**
This study evaluated 109 PMS sensors at 75 different US EPA monitoring sites with hourly FEM measurements of
$PM_{2.5}$ and/or $PM_{10}$. The US EPA provided measurements from 28 of these 109 collocated sensors, which were
previously used by Barkjohn et al. (2021). These 28 sensors are a subset of the 50 sensors originally used in the
Barkjohn paper because: 5 sensors were collocated with 24-hour averaged FRM measurements; 6 had less than 3
months of collocated measurements; 5 had poor correlation ($R^2$ less than 0.5, after removing the coarse-rich days using
FEM based coarse fraction and $PM_{10}$ concentrations); 2 were situated at beach; and 4 were already downloaded as
part of the 77 publicly available sensors (discussed below). The two sensors located at the beach were excluded due
to the high humidity and high sea salt concentrations. Of the remaining 81 sensors, 77 were publicly available sensors,
and raw data was downloaded (2-min frequency) using PurpleAir's Data Download Tool (v1.3.5), and 4 additional
PMS5003s were available from the authors' group at the University of Utah. The publicly available sensors were
considered collocated if the sensor had the same GPS coordinates (latitude and longitude) as the EPA monitoring site;
if the sensor did not have the same coordinates but was within 0.8 km, the sensor was treated as a "nearby" sensor.
Thirteen of the 77 sensors were "nearby" sensors, which increased spatial diversity by adding 13 additional monitoring
sites. This study spanned from January 2017 to May 2025; however, data availability varied by sensor, depending on
its deployment dates. The supplementary materials include sensor IDs, the corresponding collocated EPA monitoring
site IDs, and each sensor's data availability (Tables S1, S2, S3) and a map with the 75 EPA monitoring sites used in
this study (Figure S1).
The downloaded measurements included particle counts in the six size bins, RH, and pm2.5_alt. All the sensors used
in this study had a minimum of three months of reasonably continuous data. The PMS measurements were cleaned,
partially following guidelines by Barkjohn et al. (2022). Specifically, the 2-min averages were converted to hourly
measurements if 27 or more 2-minute stamps existed in an hour (>90 % completion). Otherwise, the measurement
was considered incomplete and not further analyzed. Next, the hourly measurements of dual nodes were considered
valid if (a) the difference between the pm2.5_alt values for A and B nodes of PA was less than 5 μg/m³, or (b) the
relative percentage difference was less than 61%. Barkjohn et al. (2022) used pm2.5_cf_1 ($PM_{2.5}$ mass concentration
reported by PMS sensor using a correction factor = 1), while this study used the pm2.5_alt to clean the PM
measurements. We selected pm2.5_alt because pm2.5_cf_1 can exhibit random elevated values (order of 1000s)
(Barkjohn et al., 2021), even when the number counts in the six bins are in a reasonable range. The pm2.5_alt is
calculated directly from the bin counts (Wallace, 2023) and is less susceptible to random spikes. This study also used
the PMS sensor's reported RH (using BME280, Bosch Sensortec, Germany) measurements. Therefore, PM
measurements with missing RH were excluded from the study. This resulted in the removal of <5% of the measurement
for 94 sensors, between 5–10 % for 7 sensors, between 10–22% for 6 sensors, and 34.7% and 64.9% for the CA15
and CO3 sensors.

**2.3 FEM measurements**
FEM measurements of $PM_{2.5}$ and $PM_{10}$ were accessed from the AQS site (Download Files | AirData | US EPA, 2025)
for the period between Jan 2017–July 2024 (this data was unavailable after July 2024). For the remaining period, i.e.,
between July 2024–May 2025, the data was accessed through the AirNow API (AirNow API Documentation, 2025).
Most sites employed the beta attenuation and broadband spectroscopy method (i.e., Teledyne T640 and T640x) for
Federal Equivalent Method (FEM) hourly $PM_{2.5}$ and $PM_{10}$ measurements. A few sites also used FDMS (filter dynamic
measurement system) in conjunction with a TEOM (tapered element oscillating microbalance) and laser light
scattering (GRIMM) for hourly PM measurements. Table S1, Table S2, and Table S3 detail the methods used for $PM_{2.5}$
and $PM_{10}$ measurement at each site.
Some sites had multiple parameter occurrence codes (POCs), either from different measurement methods or from
multiple instruments using the same method operating concurrently. For sites with two different methods active
simultaneously, measurements from the method with the greater number of measurements was used. For sites equipped
with the Teledyne T640X and Teledyne T640, this study used the POC corresponding to the EPA-corrected
measurements (Regulations.gov, 2025).
The FEM measurements were used in two ways in this study. First, they were used to calculate the coarse fraction
(CF), which was subsequently used to assess the effectiveness of the proposed method (Section 2.5) in identifying
periods with a high proportion of coarse particles. Dust-dominated conditions are typically characterized by high CF
values (>0.7)(Sugimoto et al., 2016). Second, they were used to evaluate the performance of the derived corrections.
When both FEM $PM_{2.5}$ and $PM_{10}$ measurements were available, the coarse fraction (CF) was calculated as:
$$CF = \frac{PM_{10} - PM_{2.5}}{PM_{10}} \tag{1}$$
Where $PM_{10}$ and $PM_{2.5}$ were concentrations in $\mu g/m^3$. Of the 109 sensors evaluated in this study, 30 sensors did not
have CF data for their evaluation period, and 36 sensors had CF data for 90% of their evaluation period. A total of 35
sensors had CF data for 40–90% of their evaluation period, and the remaining 8 sensors had CF data for less than 30%
of their evaluation period.

**2.4 Parameters**
This study used the following parameters developed from the PMS sensors to identify potential PM measurements
dominated by dust:
1. Ratio of mass in the coarse fraction to submicron fraction (C_to_SM): This ratio was defined as:
$$C\_to\_SM = \frac{\overline{M_{2.5-5}} + \overline{M_{5-10}}}{\overline{M_{0.3-0.5}} + \overline{M_{0.5-1}}} \tag{2}$$

Where $M_{i-j}$ represents the mass concentration (μg/m³) provided by the PMS sensor in the bin with size bin i–j
μm. The $\overline{M_{i-j}}$ is the average of valid $M_{i-j}$ measurements from node A and node B of the PMS sensors. The $M_{i-j}$,
used here and previously by Wallace et al. (Wallace, 2023), was calculated as:
$$M_{i-j} = \frac{4}{3}\pi \left(\frac{\sqrt{i*j}}{2}\right)^3 * N_{i-j} * 10^{-2} \qquad\qquad (3)$$
The $N_{i-j}$ represents the number counts of particles (#/dl) in the size bin i–j μm. The density was assumed to be 1
g/cm³, although density cancels out in the C_to_SM calculation. The $10^{-2}$ accounts for the unit conversions in
the equation (μm³ to m³; g/cm³ to μg/m³; and 1/dl to 1/m³).
Because the particle counts in the 2.5–5 μm and 5–10 μm size bins are much lower than in smaller bins, mass
concentrations were used instead of number concentrations to obtain more stable and interpretable ratios. The
C_to_SM parameter was used to identify potential PM measurements dominated by dust. This approach builds on
the rationale presented in Ouimette et al. (2024) and discussed in the Introduction. Briefly, although coarse PM
has a low probability of correct classification, the coarse bin will register elevated counts during dust-dominated
PM measurements. Consequently, the C_to_SM ratio becomes elevated during dust events.
2.   RH: PMS sensor RH measurements are biased low by approximately 10%–20% (Mathieu-Campbell et al., 2024).
This bias tended to increase at higher RH, although the PA's RH measurements generally show good correlation
with regulatory RH measurements ($R^2 > 0.9$) (Mathieu-Campbell et al., 2024). Dust events are typically associated
with low RHs (< 40–60%) (Csavina et al., 2014), as higher humidity tends to inhibit dust suspension and promotes
faster resettling of particles. We used an RH of 50%, as measured by the PMS sensor, as a threshold for detecting
dust. It should be noted that an RH of 50% reported by the PMS sensor corresponds to an actual RH of ~70%.
The cutoff of 50% is supported by Figure S2, which illustrates that elevated C_to_SM with high CF was
predominantly associated with RH less than 50%. This study also explored dust-dominated PM measurements
identified without the use of RH, and the results are discussed in the supplementary section S1.

**2.5 Methods for identifying potential dust-dominated PM measurements**

Our method included two approaches. The first focuses on post-processing the sensor measurements to identify
potential dust-dominated PM measurements and to develop appropriate corrections for subsequent applications. The
second approach emphasizes real-time identification of potential dust-dominated PM measurements. Both approaches
use the parameters C_to_SM and RH. The post-processing approach establishes the framework for real-time detection,
as it provides a clearer way to illustrate the methodology. However, the same real-time approach could also be applied
to post-processing the sensor data.

**2.5.1 Post-processing approach**

To identify potential dust-dominated PM measurements, two sensor-specific C_to_SM based thresholds (threshold1
and threshold2) were defined using the full dataset for each sensor.
•  Threshold1 was calculated as the sum of the median of C_to_SM and a factor (F of 2.5) times the median
absolute deviation (MAD) of the C_to_SM.
$\text{Threshold1} = \text{Median}(\text{C\_to\_SM}) + F \times \text{MAD}(\text{C\_to\_SM})$         (4)
$\text{MAD} = \text{Median}(|\text{C\_to\_SM} - \text{Median}(\text{C\_to\_SM})|)$         (5)
•  Threshold2 was defined as the maximum of:

I.   Three times the slope (with the intercept fixed at zero) from a linear regression of C_to_SM (y-axis) against pm2.5_alt (x-axis), or

II.  A value of 0.584, i.e., three times the median slope of the slopes from all the sensors evaluated in this study.

Figure 1 displays Threshold1 and Threshold2 for a subset of representative sensors (for selected sensors with a history of windblown dust impacts). Threshold1 was primarily used to differentiate clean days from those with elevated coarse particle concentrations. Because dust-dominated PM typically occurs under specific meteorological conditions (e.g., during dust events or wildfires), most measurements were expected to reflect low coarse PM concentrations. Consequently, most C_to_SM values represent these low-coarse concentration conditions, and the overall median serves as a baseline C_to_SM for such conditions. The outliers in the C_to_SM would represent high coarse concentration measurements. To identify these outliers, we excluded the measurements near the baseline by setting a threshold, i.e., 2.5 times the MAD of the median. Previous studies have reported that F values of 3, 2.5, or 2 are effective for detecting outliers (Leys et al., 2013), with $F = 3$ considered conservative and $F = 2.5$ moderately conservative.

Threshold2 helped differentiate between potential dust-dominated PM from other sources that may also increase C_to_SM. For example, during wildfires, both $PM_{10}$ and $PM_{2.5}$ levels are typically elevated, which could lead to high C_to_SM values, but also high pm2.5_alt concentrations. Threshold2, defined by the slope between pm2.5_alt and C_to_SM (Figure 1), serves as a threshold to filter out high C_to_SM values that are not associated with dust-dominated PM, specifically when both C_to_SM and pm2.5_alt are high.

Threshold1 and threshold2 were calculated using all available measurements, irrespective of the availability of FEM $PM_{2.5}$ concentrations. A measurement was labeled as a potential dust-dominated PM measurement if:

- C_to_SM exceeded threshold1, and
- C_to_SM/pm2.5_alt exceeded threshold2, and
- RH from the PA measured less than 50%

## 2.5.2 Real-time detection of the dust-dominated PM measurements

For real-time potential dust-dominated PM measurements detection, threshold1 was defined as the median of C_to_SM plus 2.5 times its MAD, calculated from the preceding 336 hourly measurements (14 days). Thus, threshold1 was dynamic. This 14-day window was selected to match the temporal coverage of the PA real-time map, which provides the most recent 14 days of hourly measurements.

Threshold2 was set at a fixed value of 0.584, from the measurements discussed in the post-processing approach (Section 2.5.1). This fixed threshold of 0.584 was selected to avoid using a slope calculated from just 336 points, which can be highly sensitive to outliers, as a few extreme values can distort the slope.

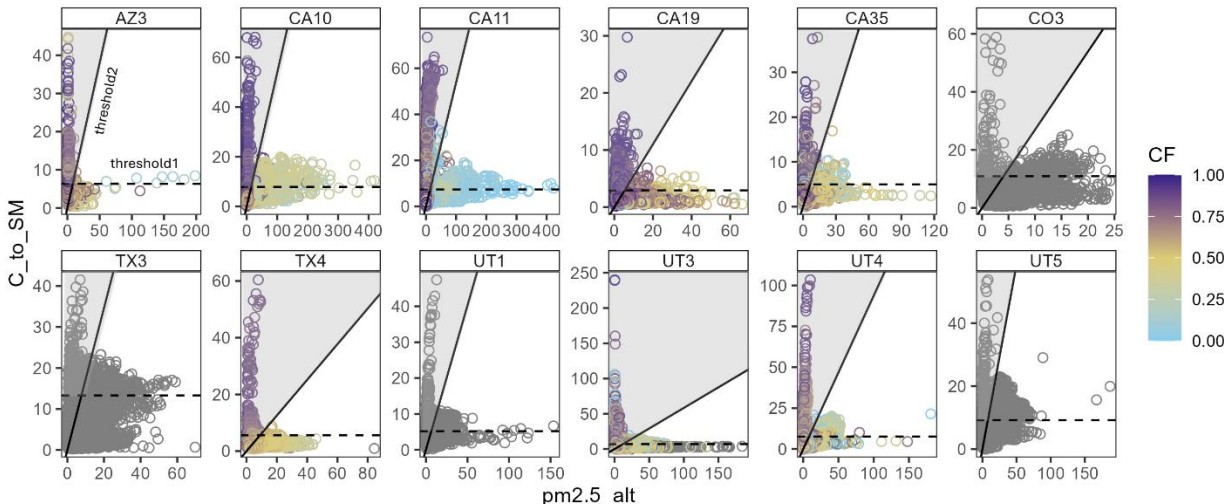

288

**Figure 1:** pm2.5_alt vs. C_to_SM. Each point represents an hourly averaged sensor measurement, colored by the FEM-calculated CF. The black dashed line represents threshold1, and the black solid line represents threshold2. The shaded grey region indicates measurements identified as potential dust-dominated PM measurements. Grey circles represent times when either $PM_{2.5}$ or $PM_{10}$ was unavailable, preventing the calculation of CF. The comparison of pm2.5_alt vs. C_to_SM for the remaining sensors is shown in Figure S3.

### 2.5.3 Evaluating the effectiveness of the two approaches in selecting dust-dominant PM conditions

Figure 1 and Figure S3 show that high C_to_SM tended to occur at low pm2.5_alt values, and measurements identified as potentially dust-dominant using threshold1 and threshold2 generally corresponded to elevated CF values. We evaluated the effectiveness of our two approaches by examining how the potential dust-dominant measurements corresponded to various bins of FEM CF values (0–0.25, 0.25–0.5, 0.5–0.7, and >0.7). High CF values can occur under low $PM_{10}$ conditions, when small PM concentrations lead to increased variability in the ratio and do not necessarily indicate dust-dominant conditions. We established criteria for "true dust-dominant PM", defined when CF >0.7 and FEM $PM_{10}$ concentrations >100 µg/m$^3$ and compared how PMS-derived $PM_{2.5}$ (pm2.5_alt) compares with FEM $PM_{2.5}$ concentrations under true dust-dominant conditions. Note that a universally accepted definition of dust-dominant conditions does not yet exist.

Even during periods classified as true dust-dominant, PMS sensors can estimate $PM_{2.5}$ reasonably well, as shown in Figure S4, which compares PMS-derived $PM_{2.5}$ (pm2.5_alt) with FEM $PM_{2.5}$ concentrations. Agreement between PMS and FEM measurements under these conditions likely reflects the presence of mixed aerosol conditions, in which coarse dust particles coexist with submicron aerosols that contribute to $PM_{2.5}$ and are detectable by the PMS sensor. The primary objective of the thresholds defined here is to identify conditions under which the PMS sensor underestimates $PM_{2.5}$. Accordingly, $PM_{2.5}$ underestimation was defined as measurements for which pm2.5_alt was less than 0.25 times the corresponding FEM $PM_{2.5}$ concentration. For the true dust-dominant periods (CF >0.7 and $PM_{10}$ >100 µg/m$^3$), the analysis quantified the fraction of measurements exhibiting $PM_{2.5}$ underestimation that were correctly identified by the thresholds. Of the 109 sensors evaluated, 79 sensors had sufficient data to calculate CF and were included in this analysis.

**2.5.4 Correction of measurements identified as dust-dominated PM measurements**

The pm2.5_alt concentration was corrected (referred to as adj_pm2.5_alt) using a non-linear regression model in R (4.4.0) that incorporated pm2.5_alt, C_to_SM, threshold1, and a scaling factor A. Measurements identified as potential dust-dominated PM measurements based on threshold1, threshold2, and RH thresholds were combined across all sensors for the post-processing approach and real-time approach. These pooled measurements were then used to estimate factor A for each approach through non-linear regression, as follows:

$$\text{adj\_pm2.5\_alt} \sim \text{FEM PM}_{2.5} = \frac{C\_to\_SM}{threshold1 * A} * \text{pm2.5\_alt} \tag{6}$$

Threshold2 was not included in the correction equation because its sole purpose was to exclude measurements with a low CF that produced elevated C_to_SM values. In contrast, threshold1 was applied to normalize C_to_SM values, enabling measurements from all sensors to be pooled together. This normalization ensured that the correction was not disproportionately influenced by sensors with high C_to_SM values. By normalizing with threshold1, data from all sensors could be combined to derive a single factor (A).

**2.6 Analysis**

Data analysis was performed using R (4.4.0). The primary focus of the analysis was to compare pm2.5_alt with FEM PM$_{2.5}$ concentrations, with an emphasis on potential dust-dominated PM measurements, and to evaluate the effectiveness of the applied correction approach. No additional corrections (i.e., adjustments for RH or FEM instrument calibration) were applied to pm2.5_alt. This study focused solely on the comparison between pm2.5_alt and FEM PM$_{2.5}$, and adj_pm2.5_alt and FEM PM$_{2.5}$. The performance of our correction approach was evaluated using the difference in the mean bias error (MBE, μg/m³) and root mean square error (RMSE, μg/m³; normalized RMSE represented as nRMSE) before and after correction of pm2.5_alt.

$$\text{MBE} = \frac{1}{n}\sum_{i=1}^{n}(\text{pm2.5\_alt}_i - \text{FEM PM2.5}_i) \tag{7}$$

$$\text{MBE\_adj} = \frac{1}{n}\sum_{i=1}^{n}(\text{adj\_pm2.5\_alt}_i - \text{FEM PM2.5}_i) \tag{8}$$

$$\text{RMSE} = \sqrt{\frac{1}{n}\sum_{i=1}^{n}(\text{pm2.5\_alt}_i - \text{FEM PM2.5}_i)^2} \tag{9}$$

$$\text{RMSE\_adj} = \sqrt{\frac{1}{n}\sum_{i=1}^{n}(\text{adj\_pm2.5\_alt}_i - \text{FEM PM2.5}_i)^2} \tag{10}$$

$$\Delta\text{RMSE} = \text{RMSE\_adj} - \text{RMSE} \tag{11}$$

$$\text{nRMSE} = \frac{\text{RMSE}}{\overline{\text{PM}_{2.5}}} \tag{12}$$

$$\text{nRMSE\_adj} = \frac{\text{RMSE\_adj}}{\overline{\text{PM}_{2.5}}} \tag{13}$$

MBE, MBE_adj, RMSE, RMSE_adj, nRMSE, nRMSE_adj, and ΔRMSE were calculated only for those measurements identified as potential dust-dominated PM measurements; the remaining measurements were not corrected and not included in the calculation. Measurements identified using the real-time approach were corrected using three different values of A: (1) A derived from a non-linear regression (Eq. 6) using all measurements identified by the real-time approach; (2) A derived from a non-linear regression (Eq. 6) using measurements identified by the post-processing approach; and (3) sensor-specific A values, derived from sensor-specific non-linear regressions (Eq. 6) using real-time measurements for each sensor.

In the main manuscript, we present results for 12 representative sensors that have collocated measurements and are affected by windblown dust, including sensors in Utah, Arizona, Texas, Colorado, and California. The results for the remaining locations are discussed in the supplementary material.

**3 Results and discussion**

**3.1 Measurements identified as a potential dust-dominated PM using the thresholds**

Table S4 summarizes the counts of the threshold-based identified measurements, which identified 0 – 3785 (post-processing approach: 0 – 9.00 % of the sensor's total hourly measurements) and 0 – 4513 (real-time approach: 0 – 9.62 % of the sensor's total hourly measurements) hourly measurements as potential dust-dominated PM measurements. These counts did not consider the availability of FEM $PM_{2.5}$ measurements. The real-time approach generally identified more measurements than the post-processing approach (Figure S5, Table S4). This outcome was expected because the real-time method used a dynamic threshold1, whereas the post-processing method used a constant threshold1. A dynamic threshold1 accommodated shifts in the C_to_SM baseline (defined in Section 2.5.1), which can occur when a sensor operates for extended periods (example in Figure S6) or when a PMS sensor is replaced within a node (example in Figure S6), resulting in C_to_SM baseline shifts due to differences in sensor-specific performance characteristics.

Seasonal variability in C_to_SM further complicated the use of a constant threshold1 (example in Figure S6). When most measurements originated from seasons with elevated PM concentrations, the overall threshold1 was biased upward, leading to the rejection of high-coarse concentration measurements during seasons with lower C_to_SM values. The reverse was held when measurements were dominated by low-concentration seasons. In contrast, the dynamic threshold1 adjusted for these seasonal shifts, thereby improving the ability to identify dust-dominated PM measurements (Figure S6).

Figure S7 compares the number of PMS measurements identified as potential dust-dominated PM measurements to the CF, grouped by CF bins (0–0.25, 0.25–0.5, 0.5–0.7, and >0.7). The majority of potential dust-dominated measurements were associated with the CF > 0.7 bin ($67.1 \pm 23.4\%$ for the post-processing approach and $70.2 \pm 20.8\%$ for the real-time approach), followed by the bin between 0.5 and 0.7 ($18.9 \pm 17.1\%$ and $19.2 \pm 15.9\%$, respectively). Fewer than 8.50% of measurements fell within the 0.25–0.5 bin, and fewer than 5.50% fell within the 0–0.25 bin. This distribution of potential dust-dominated PM measurements, with most measurements in CF > 0.7, supported the use of the thresholds derived in this study. The CF between 0.5 and 0.7 could have represented dust mixed with other sources.

Tables 1 and S5 summarize counts of true dust-dominant PM measurements and measurements with PMS $PM_{2.5}$ underestimation across all sensors. For the 12 representative sensors (CF available for 8 sensors, Table 1), approximately 20% of measurements identified using our thresholds were true dust-dominant PM measurements, and roughly 16% were underestimated (pm2.5_alt / $PM_{2.5}$ < 0.25). The post-processing and real-time approaches captured $68.4 \pm 24.7\%$ and $72.6 \pm 22.2\%$, respectively, of all measurements corresponding to true dust-dominated PM measurements. When restricted to underestimated PMS $PM_{2.5}$ measurements meeting the same criteria, $86.5 \pm 14.9\%$ (post-processing) and $89.7 \pm 9.9\%$ (real-time) were captured. Across the remaining 71 sensors (Table S5), 11–12% of measurements identified using our thresholds were true dust-dominant PM measurements, and 7–10% exhibited PMS $PM_{2.5}$ underestimation. Although these remaining 71 sensors had a smaller fraction of true dust-dominant PM, they accounted for $60.6 \pm 30\%$ and $66.7 \pm 25.6\%$ of the underestimated PMS $PM_{2.5}$ measurements, in the post-processing and real-time approaches, respectively.

**Table1:** Counts of measurements with available CF; true dust-dominant measurements (CF >0.7 and $PM_{10}$ >100 µg/m³); and true dust-dominated PM measurements with underestimated PMS $PM_{2.5}$ concentration (pm2.5_alt/$PM_{2.5}$ <0.25). Counts are reported for all measurements, for measurements identified using the post-processing approach, and for measurements identified using the real-time processing approach. Four of the 12 representative sensors did not have the appropriate measurements to calculate CF. Table S6 summarizes the counts for the remaining sensors.

| Sensor Name | CF available | | | True dust-dominated PM (CF>0.7 & $PM_{10}$>100 µg/m³) | | | True dust-dominated PM & underestimated (CF>0.7 & $PM_{10}$>100 µg/m³ & pm2.5_alt/$PM_{2.5}$ <0.25) | | |
|---|---|---|---|---|---|---|---|---|---|
| | all | post-processing | real-time | all | post-processing | real-time | all | post-processing | real-time |
| AZ3 | 6713 | 472 | 581 | 70 | 40 | 41 | 47 | 37 | 37 |
| CA10 | 34892 | 1895 | 2065 | 488 | 372 | 380 | 377 | 341 | 340 |
| CA11 | 45865 | 3754 | 4461 | 898 | 817 | 861 | 823 | 771 | 798 |
| CA19 | 9642 | 552 | 590 | 709 | 281 | 295 | 251 | 207 | 208 |
| CA35 | 27470 | 683 | 779 | 354 | 97 | 151 | 151 | 82 | 112 |
| TX4 | 2132 | 190 | 140 | 33 | 32 | 32 | 30 | 30 | 30 |
| UT3 | 25609 | 1179 | 1363 | 187 | 149 | 156 | 148 | 144 | 144 |
| UT4 | 16753 | 847 | 1026 | 106 | 84 | 89 | 86 | 82 | 84 |

## 3.2 $PM_{2.5}$ vs. pm2.5_alt: post-processing approach

Figure 2 (top) compares the pm2.5_alt with FEM $PM_{2.5}$ concentrations, with measurements identified as potential dust-dominated PM marked as black circles, for the 12 representative sensors. The identified measurements corresponded predominantly to cases in which pm2.5_alt underestimated FEM $PM_{2.5}$, typically associated with CF >0.7. The potential dust-dominant measurements identified from all sensors were pooled, and using Eq. 6, factor A was estimated at 0.702. The corrections were applied to measurements identified as potentially dust-dominated PM, while all other observations remained unchanged. Figure 2 (bottom) illustrates the measurements after correction. The coefficient of determination ($R^2$, using just the identified measurements) increased from 0.290 for pm2.5_alt versus FEM $PM_{2.5}$ to 0.653 for adj_pm2.5_alt versus FEM $PM_{2.5}$. Sensor-specific changes in $R^2$ are presented in Table S6. Comparisons of pm2.5_alt and FEM $PM_{2.5}$ for the remaining sensors were presented in Figures S8 and S9.

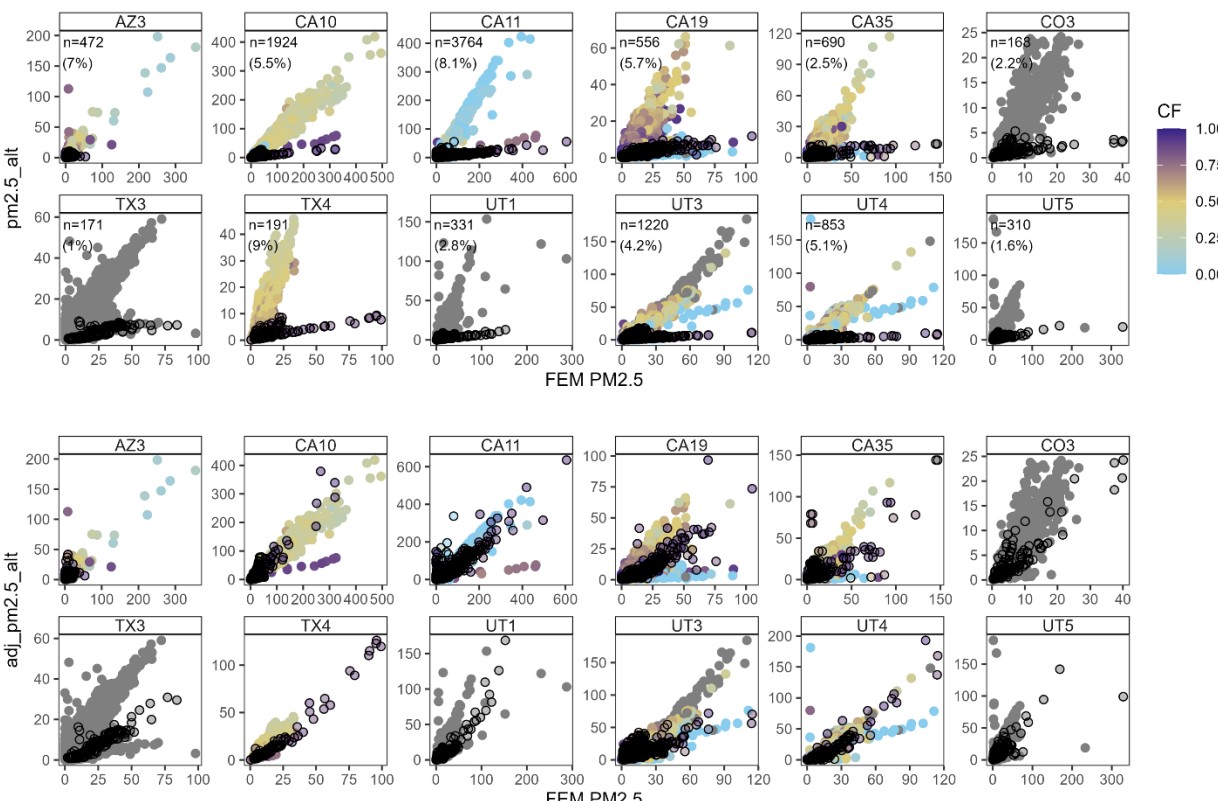

**Figure 2:** Post-processing approach: (top) Hourly averaged pm2.5_alt compared with FEM $PM_{2.5}$ measurements. Black open circles indicate measurements identified as dust-dominated PM measurements. (bottom) Hourly averaged adj_pm2.5_alt compared with FEM $PM_{2.5}$ measurements. The color of the solid circles corresponds to the CF. The grey solid circles represent times when either $PM_{2.5}$ or $PM_{10}$ was unavailable, preventing the calculation of CF. n denotes the number of measurements identified as potential dust-dominated PM measurements, for a subset of data when FEM $PM_{2.5}$ data were available. Table S4 shows the total number of potential dust-dominated PM measurements, regardless of FEM $PM_{2.5}$ availability. Comparisons of pm2.5_alt and FEM $PM_{2.5}$ for the remaining sensors were presented in Figures S8 and S9.

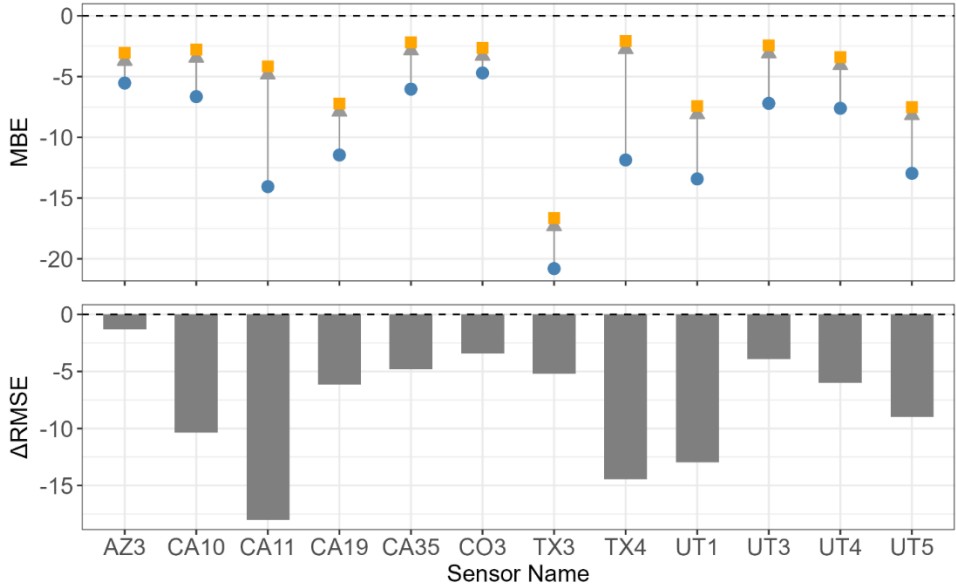

422

**Figure 3:** Post-processing approach: (top) MBE ($\mu g/m^3$) with respect to FEM $PM_{2.5}$ before and after correcting pm2.5_alt. The blue dot represents the MBE before correction, while the orange square represents the MBE after correction (MBE_adj). The grey arrow highlights the direction of the MBE shift after correction. (bottom) change in RMSE ($\Delta$RMSE, $\mu g/m^3$). Comparisons of MBE and RMSE for the remaining sensors are presented in Figure S10, Table S4, and Table S7. Table S7 also reports $\Delta$nRMSE for all the sensors.

428

Figure 3 presents MBE, MBE_adj, and $\Delta$RMSE. MBE value closer to zero indicates better sensor performance, meaning that the sensor-estimated $PM_{2.5}$ concentrations are more consistent with the FEM $PM_{2.5}$ measurements. For the 12 representative sensors shown in the main manuscript, MBE decreased by 52.4 ± 16.9% on average, with MBE ranging between -20.8 and -4.71 $\mu g/m^3$ and MBE_adj ranging between -16.7 and -2.07 $\mu g/m^3$. These results indicate that although the sensor's $PM_{2.5}$ estimation improved it remained less than FEM $PM_{2.5}$ after correction. For these 12 sensors, RMSE also decreased by 42.3 ± 15.9%, with $\Delta$RMSE ranging between -18.0 and -1.3 $\mu g/m^3$, indicating a reduction in error following correction. The corresponding nRMSE shows the same trends as those for RMSE, and these values can be found in Table S7.

For the remaining sensors (Figure S10, Table S4, and Table S7), 82 sensors showed decreases in MBE, with MBE ranging between -14.6 and -1.54 $\mu g/m^3$ to MBE_adj between -7.60 and 0.76 $\mu g/m^3$, i.e., a bias error reduction of 49.6 ± 22.9 %. The RMSE decreased by 27.2 ± 14.3 % for 72 sensors ($\Delta$RMSE ranging between -15.8 and -0.0965 $\mu g/m^3$) and increased for 23 sensors ($\Delta$RMSE ranging between 0.127 and 15.9 $\mu g/m^3$). Five sensors (Figure S10) showed a decrease in MBE (with MBE_adj ranging from -3.52 to -0.818 $\mu g/m^3$), but an increase in $\Delta$RMSE (from 11.2 to 17.5 $\mu g/m^3$). Extreme overcorrection of a few measurements for these five sensors was likely responsible for this behavior.

Thirteen sensors exhibited MBE_adj greater than 1 $\mu g/m^3$ and positive $\Delta$RMSEs (Figure S10 and Table S7), indicating the correction led to an overestimation of the PMS $PM_{2.5}$ concentrations. Among the sensors with positive MBE_adj (Figure S10), 7 had fewer than 60 potential dust-dominated PM measurements (<0.6%, Table S4), and these 7 belonged to locations in Alaska, Iowa, Oregon, Washington, and Vermont that are infrequently affected by dust. Even for sensors with positive MBE_adj, the corrected pm2.5_alt concentration remained within the sensor's expected range (Figure S9). For two sensors, no measurements were identified as potential dust-dominated PM.

It is important to note that the locations of most sensors were identified based on the PA public map, and no physical
verification of collocation was available, except for 4 sensors maintained by the authors' group and the sensor list
provided by the EPA. Given this uncertainty, some variability in MBE and RMSE outcomes were expected.

### 3.3 Real-time processing: pm2.5_alt vs. FEM PM$_{2.5}$

Figure 4 (top) compares the pm2.5_alt with FEM PM$_{2.5}$ concentrations, with measurements identified as potential
dust-dominated PM marked as black circles, for the 12 representative sensors, using the real-time approach. The
potential dust-dominant measurements identified from all sensors were pooled, and using Eq. 6, factor A was estimated
at 0.998. The corrections were applied to measurements identified as potentially dust-dominated PM, while all other
observations remained unchanged. The real-time approach identified measurements were also corrected using an A of
0.702 (derived using the post-processing approach) and sensor-specific As. Figure 4 (bottom) illustrates the
measurements after correction using an A of 0.702. The coefficient of determination (R², using just the identified
measurements), irrespective of the A values, increased from 0.262 for pm2.5_alt versus FEM PM$_{2.5}$ to 0.513 for
adj_pm2.5_alt versus FEM PM$_{2.5}$. Sensor-specific changes in R² are presented in Table S6.

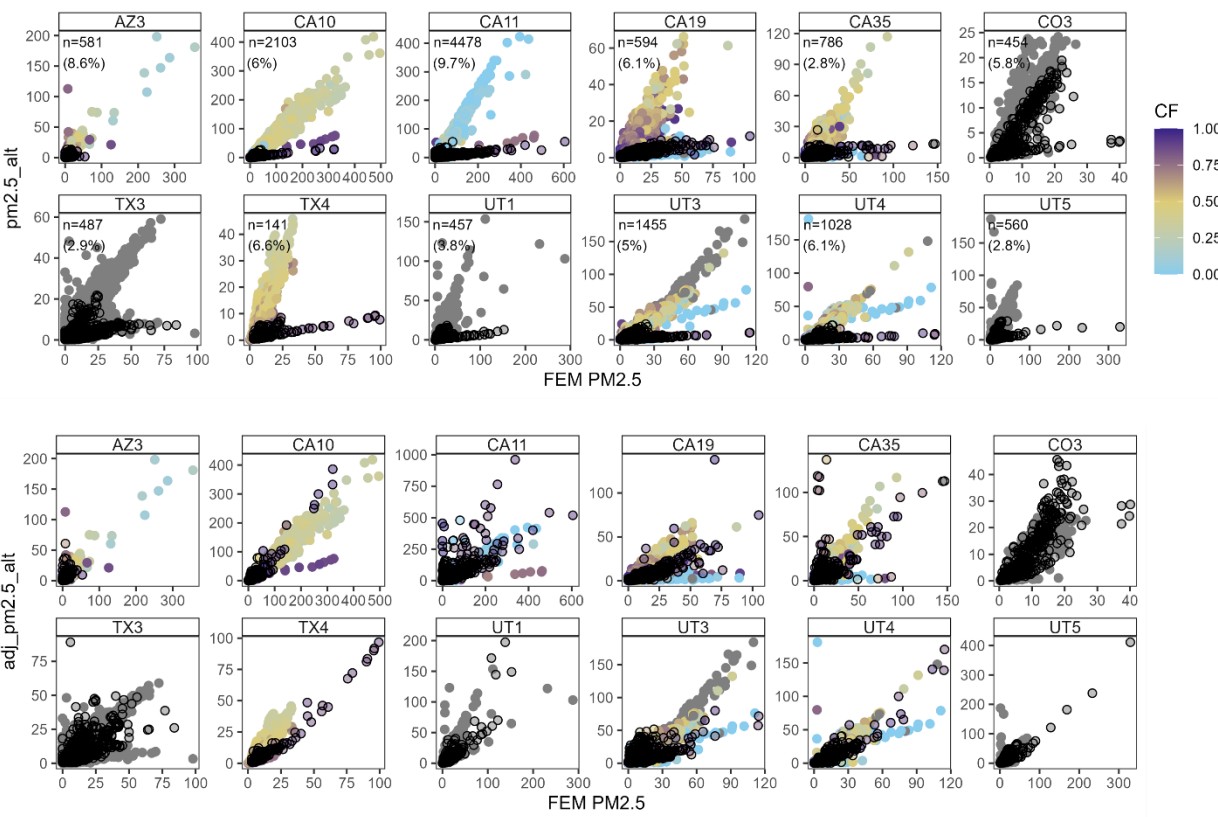



**Figure 4:** Real-time approach: (top) Hourly averaged pm2.5_alt values compared to FEM PM$_{2.5}$ measurements.
(bottom) Hourly averaged adj_pm2.5_alt, corrected using an A of 0.702, compared with FEM PM$_{2.5}$ measurements.
The color of the solid circles corresponds to the CF. The grey solid circles represent times when either PM$_{2.5}$ or PM$_{10}$
was unavailable, preventing the calculation of CF. Black open circles represent measurements identified as potential
dust-dominated PM measurements. n denotes the number of measurements identified as potential dust-dominated PM
measurements, for a subset of data when FEM PM$_{2.5}$ data were available. Table S4 shows the total number of potential
dust-dominated PM measurements, regardless of FEM PM$_{2.5}$ availability. Figure S11 shows comparisons of pm2.5_alt
and FEM PM$_{2.5}$ for the remaining sensors. Figure S12 and Figure S13 compare the adj_pm2.5_alt with FEM PM$_{2.5}$,
for A of 0.998 and sensor-specific As, respectively.

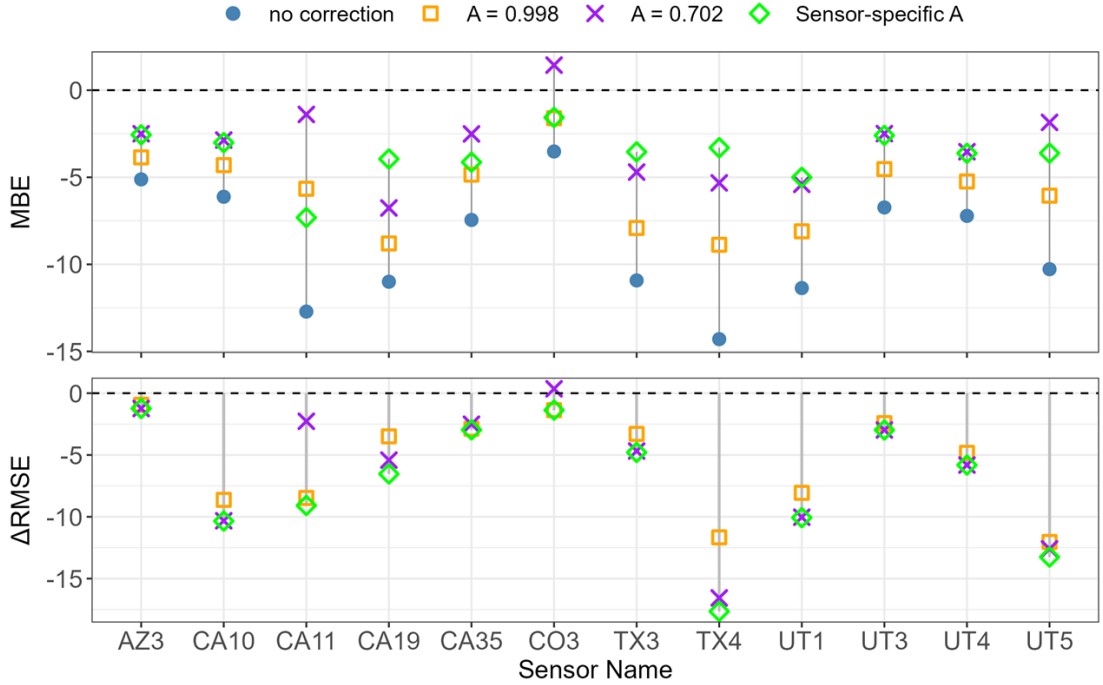


**Figure 5:** Real-time approach: (top) MBE with respect to FEM PM$_{2.5}$ before and after correcting pm2.5_alt using an
A of 0.998, 0.702, and sensor-specific A. The blue dot represents the MBE before correction, while the orange square,
purple cross, and green diamond represent the MBE after correction (MBE_adj) using an A of 0.998, 0.702, and
sensor-specific A, respectively. (bottom) change in RMSE (ΔRMSE) with respect to FEM PM$_{2.5}$ before and after
correcting pm2.5_alt using an A of 0.998, 0.702, and sensor-specific A. The orange square, purple cross, and green
diamond represent the ΔRMSE after correction using an A of 0.998, 0.702, and sensor-specific A, respectively. Figure
S14, Table S4, and Table S7 show comparisons of MBE and MBE_adj, and ΔRMSE for the remaining sensors. Table
S7 also reports ΔnRMSE for all the sensors.

Figure 5 compares the MBE and RMSE before and after correction using different values of A for the 12 representative
sensors. The results indicate that the corrected PMS PM$_{2.5}$ concentrations agreed better with the FEM PM$_{2.5}$
concentrations during periods potentially dominated by dust, regardless of the A value. The sensor-specific A and A
value of 0.998 led to decreases in MBE and RMSE and did not cause overestimates of the PM$_{2.5}$ concentration
compared to the FEM PM$_{2.5}$ concentration. Using an A of 0.702 caused one sensor (CO3) to slightly overestimate
PM$_{2.5}$ concentration compared to the FEM, with an increase in MBE from -3.52 μg/m$^3$ to 1.44 μg/m$^3$.
Figure S14, Table S4, and Table S7 presents changes in MBE and RMSE for the remaining 95 sensors (no
measurements identified with the thresholds for 2 sensors) using different values for A. Applying a correction factor
with A = 0.998 reduced MBE for 91 sensors by 24.7 ± 12.7% on average and RMSE for 80 sensors by 14.5 ± 9.96%
with ΔRMSE between -8.53 and -0.026 μg/m$^3$. This correction (A = 0.998) caused an increase in MBE for four sensors
(MBE_adj between 0.318 and 4.06 μg/m$^3$) and RMSE for 12 sensors (ΔRMSE between 0.025 and 11.0 μg/m$^3$),
suggesting inconsistent performance across sensors. Application of a correction factor with A = 0.702 resulted in MBE
decreases for 83 sensors by 55.4 ± 19.9% (with MBE_adj varying between -7.58 and -0.0035 µg/m$^3$) and RMSE
decreases for 71 sensors by 24.8 ± 14.5% (with ΔRMSE in -13.3 to -0.0422 µg/m$^3$). This correction caused increases
in MBE at 13 sensors (with MBE_adj between 0.313 and 8.76 µg/m$^3$). For 11 of these 13 sensors, RMSE also increased
(MBE_adj between 0.13 and 6.78 µg/m$^3$, and ΔRMSE between 0.11 and 17.96 µg/m$^3$, respectively), with 6 sensors
belonging to locations in Alaska, Oregon, Washington, and Vermont that are infrequently affected by dust. For a few
sensors (NV3, CA14, CA21, CO1, and CO2), a decrease in MBE was accompanied by an increase in RMSE (positive
ΔRMSE), indicating sensitivity to a small number of overcorrected observations.
The sensor-specific A correction produced the greatest improvement in performance, yielding the largest reduction in
MBE across sensors (mean reduction of 58.9 ± 23.9% across 85 sensors) and a decrease in RMSE of 28.9 ± 19.4%
for 82 sensors (Figure S14; Table S7). Although seven sensors exhibited increases in both MBE (24.9 ± 17.1%) and
RMSE (ΔRMSE = 0.32–2.61 µg/m$^3$) following correction, and six additional sensors showed reduced bias but
increased RMSE (ΔRMSE = 0.28–0.63 µg/m$^3$), these cases reflect a trade-off between improved mean agreement and
increased variability. Importantly, for sensors with increased RMSE, the magnitude of ΔRMSE using the sensor-
specific correction was substantially smaller (0.28–2.61 µg/m$^3$) than that observed under the uniform-A correction
(0.024–10.9 µg/m$^3$ for A = 0.998 and 0.111–17.9 µg/m$^3$ for A = 0.702), indicating greater robustness of the sensor-
specific approach.
Overall, these results highlight that while uniform correction factors can reduce bias for many sensors, they may
introduce overcorrection and increased error at others. The sensor-specific correction provided a more balanced
adjustment across the network, with reduced sensitivity to overcorrection and smaller increases in absolute error where
performance degrades. However, implementing sensor-specific A requires calibration against a reference instrument
before deployment, ideally under a range of PM concentrations and compositions representative of the target
environment. This requirement limits scalability and may not be feasible for geographically diverse networks. Thus,
there is a trade-off: a fixed A offers simplicity and consistency for a broad distribution of sensors, while a sensor-
specific A improves accuracy but reduces generality.

## 4 Limitations

This study has several limitations, primarily related to the use of PMS sensors and the assumptions made in selecting
and interpreting the data. First, most PMS sensors used in this analysis were identified from the publicly available PA
map, and their physical locations and deployment conditions could not be independently confirmed. It is possible that
some of the sensors selected were not truly collocated with the FEM instruments. Second, PA nodes are user-deployed
and can be moved or reconfigured at any time. A user might relocate the entire sensor, swap sensor nodes, or even
replace hardware without any indication in the metadata. Such changes can alter sensor performance or the
environmental context of the measurements (e.g., from outdoor to indoor), potentially affecting C_to_SM values and
the thresholds used for dust-dominated PM measurement detection. These untracked changes may lead to
inconsistencies in the correction approach, either causing genuine dust-dominated PM measurements to be missed or
non-dust days to be mistakenly corrected due to a sudden shift in sensor behavior. Additionally, the correction method
depends on long-term consistency in sensor performance. Any drift in sensor response (deSouza et al., 2023),
contamination of the sensor inlet, changes in the PMS production process (i.e., (Searle et al., 2023)), or firmware
updates may also influence measurement characteristics and correction effectiveness. A potential limitation of this
approach is that it may be less effective under consistently high-dust conditions, as the baseline correction assumes
that the environment is relatively clean most of the time. Finally, this study evaluated PMS5003/6003s, and the
proposed methods would need to be evaluated for other PMS models. Despite these limitations, the general trends and
methodology proposed in this manuscript can provide a useful framework for real-time and retrospective identification
of possible dust-dominated PM measurement using PMS sensors. However, future work should aim to validate sensor-
reference collocation and investigate the impact of node-level changes on the robustness of corrections.
Some potentially problematic measurements were not explicitly excluded in this study. These included: (i) periods
with all zero counts in bins >0.5 μm throughout the sampling duration; (ii) spurious temperature readings (e.g., ~ −
129 °F (~ -89 °C)) persisting over the study period; and (iii) inconsistent particle count assignments, such as higher
counts in the >0.5 μm bin compared to the >0.3 μm bin, or in the >2.5 μm bin compared to the >5 μm bin, which
sometimes resulted in negative pm2.5_alt concentrations. While the thresholds developed in this study may have
excluded many of these problematic data points, they may also have inadvertently excluded valid dust-dominated PM
measurements.

**5 Future Work**
Our approach for identifying elevated coarse particle concentrations could be extended to improve $PM_{10}$ estimation
from low-cost sensors. Furthermore, in conjunction with back-trajectory models, meteorological data, or satellite
imagery, this method could help identify the sources of $PM_{10}$ plumes, such as dust sources, construction activity, or
agricultural emissions. In addition, if A could be defined for each sensor based on laboratory calibration under
controlled conditions, it could significantly enhance the accuracy of real-time dust-dominated PM measurement
detection and correction, although a strategy for addressing sensor performance changes over time would still be
needed. This could enhance both the scalability and robustness of using low-cost sensors, such as the PMS, for dust-
dominated PM monitoring in diverse environmental settings.

**6 Conclusion**
This study demonstrates that PMS5003/6003 sensors, despite their well-known limitations in detecting coarse
particles, can be used to identify and provide estimates of $PM_{2.5}$ concentration during dust-dominated periods using
only the sensor's reported outputs. By leveraging particle counts in the coarser and submicron bins and RH, we
developed real-time thresholds (threshold1 and threshold2) that can identify potential dust-dominated PM
measurements without reliance on external datasets. Between 0 and 3785 hourly averaged $PM_{2.5}$ measurements (0 –
9.00%) from each sensor were identified as potential dust-dominated PM measurements with the post-processing
approach, and 0 – 4513 measurements (0 – 9.62%) with the real-time approach. The real-time method consistently
identified more dust-dominated PM measurements, owing to its dynamic threshold1, which better accounted for
seasonal and sensor-specific variability. Most potential dust-dominated PM measurements were associated with coarse
fraction values >0.7 (67–70%) as measured by FEMs, confirming that the thresholds targeted conditions when PMS
sensors most strongly underestimated FEM $PM_{2.5}$. The correction of PMS $PM_{2.5}$ estimates (pm2.5_alt) using the post-
processing approach reduced MBE by approximately 50% (52.4 ± 16.9% for the 12 representative sensors and 49.6 ±
22.9% for the remaining 85 sensors) and decreased RMSE for 84 sensors (−18.0 to −0.0965 μg/m³). A small subset of
sensors (≤13) exhibited increases in MBE and RMSE, likely due to overcorrection or limited dust-influenced
observations; however, corrected concentrations remained within the expected sensor response range. For the real-
time correction approach, the magnitude of adjustment depended on the value of A, with uniform corrections using A
= 0.998 and A = 0.702 reducing MBE and RMSE for most sensors (e.g., A = 0.998 reduced MBE by 24.7 ± 12.7% for
91 sensors, and A = 0.702 reduced MBE by 55.4 ± 19.9% for 83 sensors), although overcorrection occurred for a
limited number of sensors, resulting in increases in MBE (up to 8.76 μg/m³) and RMSE (up to 17.96 μg/m³). The
sensor-specific correction yielded the greatest overall improvement, reducing MBE by 58.9 ± 23.9% across 85 sensors
and RMSE by 28.9 ± 19.4% for 82 sensors, suggesting the most robust agreement with FEM $PM_{2.5}$ during dust-
influenced periods. Overall, the framework developed here improves PMS5003 performance under elevated dust
loading, reduces $PM_{2.5}$ underestimation, and enhances the utility of low-cost sensors for dust monitoring in regions
with limited FRM/FEM coverage.


## 7 Acknowledgement

Research reported in this publication is based upon work supported by the National Science Foundation under grant no. 2012091 (Collaborative Research Network Cluster: Dust in the Critical Zone) and under grant no. 2322009 (CIVIC-Stage2: TRACK A: Community Resilience through Engaging, Actionable, Timely, High-Resolution Air Quality Information – CREATE-AQI). The authors wish to thank Dr. Karoline Barkjohn (US EPA) for the valuable feedback on the manuscript and for providing a collocation dataset of PA sensors and regulatory monitors. This EPA collocation dataset would not have been possible without the partnership of many air monitoring agencies and other partners, including the State of Alaska, Citizens for Clean Air (Alaska), Maricopa County Air Quality Department, San Luis Obispo County Air Pollution Control District, Mojave Desert Air Quality Management District, California Air Resources Board, Santa Barbara County Air Pollution Control District, Ventura County Air Pollution Control District, South Coast Air Quality Management District and its Air Quality Sensor Performance Evaluation Center (AQ-SPEC), Colorado Department of Public Health and Environment, Delaware Division of Air Quality, Sarasota County Government, Georgia Environmental Protection Division, Iowa Department of Natural Resources, Kansas Department of Health and Environment, Missoula County, Montana Department of Environmental Quality, Forsyth County Office of Environmental Assistance and Protection, Clean Air Carolina, Quapaw Nation, Oklahoma Department of Environmental Quality, Virginia Department of Environmental Quality, State of Vermont, Puget Sound Clean Air Agency, and Wisconsin Department of Natural Resources. Thanks to PurpleAir for allowing access to their data at no cost. AI-assisted tools were used to edit the original drafts of the manuscript to enhance the clarity and readability of the text.

## 8 Conflict of interest

Kerry E. Kelly has a financial interest in the company Tellus Networked Solutions, LCC, which commercializes solutions for environmental monitoring. Their technology was not used as part of this work.

## 9 Author contribution

KK and TM gathered data. KK conceptualized the research and analyzed the data. KK developed the original draft, and TM and KEK reviewed and edited the original draft. KEK provided supervision and acquired funding.

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
