# Peer review of "Correction of PM2.5 Underestimation in Low-Cost Sensors under 2 Elevated Dust Loading Using Only Sensor Measurements"

_EGUsphere, 2025_

## Author Response (AR1)

**Reviewer1**

1. This paper investigates an important limitation of existing low-cost sensors for providing reliable data during dust-impacted events, and proposes an approach to both identify these events and compensate for sensor underestimation of $PM_{2.5}$ during identified events. The proposed approach is potentially generalizable to many currently deployed low-cost sensors using the PMS sensing technology, thereby enhancing the value of the data provided by these sensors. Currently, the manuscript lacks a thorough investigation of the performance of the proposed methods of dust-impacted measurement detection, instead focusing on the calibration approach applied to these measurements. Prior to this, assessment of the accuracy of the identification approaches should be carried out by means of common classification metrics, such as false positive and false negative rates, precision (ratio of true positives to all positives), and recall (ratio of true positives to all actual events, as determined by the FEM-derived CF). While Lines 326-332 discuss the relative fraction of detected events corresponding to different CF bins, it is not clear from this discussion what fraction of dust-impacted events (as determined by the FEM-derived CF) were correctly identified by the detection algorithm, or what fraction of events identified by the detection algorithms corresponded to high-CF events. This is a critical aspect of the analysis which is currently overlooked.

Response: We thank the reviewer for the comments. Our original title of the manuscript was misleading, and we have changed the title to "*Correction of PM2.5 Underestimation in Low-Cost Sensors under Elevated Dust Loading Using Only Sensor Measurements*".

The objective of the manuscript was not to identify the dust-events, which would require FEM PM measurements along with meteorological parameters, but rather to identify periods with elevated dust loading, which may or may not occur during dust events. We have added the following text under section 2 to further clarify:

"*This study focuses on identifying conditions that are specifically associated with PMS underestimation. It does not attempt to identify "dust events" in part because there is no well-defined dust event classification method based solely on FRM PM10 and PM2.5 measurements. Moreover, the PMS sensor's performance depends strongly on the underlying particle size distribution (Kaur and Kelly, 2023a; Kuula et al., 2020; Ouimette et al., 2024). Although the PMS sensors are inefficient at measuring particles with diameters > 1 μm (Kaur and Kelly, 2023a; Kuula et al., 2020; Ouimette et al., 2024), the PMS sensor can still provide reasonable estimates of PM2.5 concentrations when concentrations of particles > 1 μm in diameter are elevated, as well as particles < 1 μm in diameter.*"

We have also replaced the term "*dust events*" with "*dust-dominated PM*" throughout the manuscript.

We also appreciate the reviewer's suggestions about evaluating the fraction of correctly identified dust events, and we added a section about the fraction of selected measurements that actually correspond to dust-dominated periods under sections 2.5.3 and 3.1.

*"2.5.3 Evaluating the effectiveness of the two approaches in selecting dust-dominant PM conditions*

*Figure 1 and Figure S3 show that high C_to_SM tended to occur at low pm2.5_alt values, and measurements identified as potentially dust dominant using threshold1 and threshold2 generally corresponded to elevated CF values. We evaluated the effectiveness of our two approaches by examining how the potential-dust dominant conditions corresponded to various bins of FEM CF values (0–0.25, 0.25–0.5, 0.5–0.7, and >0.7). High CF values can occur under low PM10 conditions, when small PM concentrations lead to increased variability in the ratio and do not necessarily indicate dust-dominant conditions. We established criteria for "true dust-dominant PM", defined when CF > 0.7 and FEM PM10 concentrations > 100 µg/m3 and compared how PMS-derived PM2.5 (pm2.5_alt) compare with FEM PM2.5 concentrations under true dust-dominant conditions. Note that a universally accepted definition of dust-dominant conditions does not yet exist.*

*Even during periods classified as true dust events, PMS sensors can estimate PM2.5 reasonably well, as shown in Figure S4, which compares PMS-derived PM2.5 (pm2.5_alt) with FEM PM2.5 concentrations. Agreement between PMS and FEM measurements under these conditions likely reflects the presence of mixed aerosol conditions, in which coarse dust particles coexist with submicron aerosols that contribute to PM2.5 and are detectable by the PMS sensor. The primary objective of the thresholds defined here is to identify conditions under which the PMS sensor underestimates PM2.5. Accordingly, PM2.5 underestimation was defined as measurements for which pm2.5_alt was less than 0.25 times the corresponding FEM PM2.5 concentration. For the true dust-dominant periods (CF > 0.7 and PM10 > 100 µg/m3), the analysis quantified the fraction of measurements exhibiting PM2.5 underestimation that were correctly identified by the thresholds. Of the 109 sensors evaluated, 79 sensors had sufficient data to calculate CF and were included in this analysis."*

Discussion under section 3.1:

*"Tables 1 and S5 summarize counts of true dust-dominant PM measurements and measurements with PMS PM2.5 underestimation across all sensors. For the 12 representative sensors (CF available for 8 sensors, Table 1), approximately 20% of measurements identified using our thresholds were true dust-dominant PM measurements, and roughly 16% were underestimated (pm2.5_alt / PM2.5 < 0.25). The post-processing and real-time approaches captured 68.4 ± 24.7% and 72.6 ± 22.2%, respectively, of all measurements corresponding to true dust-dominated PM measurements. When restricted to underestimated PMS PM2.5 measurements meeting the same criteria, 86.5 ± 14.9% (post-processing) and 89.7 ± 9.9% (real-time) were captured. Across the remaining 71 sensors (Table S5), 11–12% of measurements identified using our thresholds were true dust-dominant PM measurements, and 7–10% exhibited PMS PM2.5 underestimation. Although these remaining 71 sensors had a smaller fraction of true dust-dominant PM, they accounted for 60.6 ± 30% and 66.7 ± 25.6% of the underestimated PMS PM2.5 measurements, in the post-processing and real-time approaches, respectively."*

Besides this major point, and the other specific comments listed below, I believe that the paper can be suitable for publication after revision. Specific Comments

2. Lines 58-59: This is a continuation of the point made in lines 48-49; these can be combined.

Response: Agreed, we have removed line 58-59.

3. Line 103: This is a sentence fragment.

Response: Thank you for noticing this error, we have corrected it.

4. Lines 134-136: Please clarify what is meant by "alternate PMS5003" as compared to PMS5003, and why the ratio of these quantities is relevant here.

Response: We have added the description: "*The ratio of >0.5 um to 0.3 um (ratio greater than 0.4) was used to identify the alternate PMS5003, i.e., a new PMS5003 version appeared in June 2021 and exhibited PM2.5 concentrations that were biased low (Searle et al., 2023)*"

5. Line 142: "are subset" should be "are a subset".

Response: Agreed, we have made the suggested changes.

6. Line 143: "month" should be "months"

Response: Agreed, we have made the suggested changes

7. Line 145: Clarify why "at beach" disqualified these sensors.

Response: We have added "The sensors at the beach were excluded due to persistent high humidity and high sea salt concentrations. "

8. Line 145: If 4 sensors were included in the study, they should not be counted among the non-included sensors.

Response: We agree that our language was unclear. The 4 sensors were not included from the EPA download of measurements from the 50 sensors, but they were included in this study. The measurements from these 4 sensors were already downloaded by the authors, and we used this downloaded data instead of using the EPA provided data because it covered a slightly longer time period. We changed the language to: "*4 were already downloaded as part of the 77 publicly available sensors (discussed below).* "

9. Line 155: It is mentioned later that T is also included, though it does not seem to be used elsewhere in the analysis.

Response: We thank the reviewer for noticing this inconsistency. Temperature (T) was not used in the analysis. We have removed the T from the manuscript.

10. Line 202: Can you explain why mass concentrations were used here, rather than the number counts directly?

Response: The mass concentrations were used to amplify the ratio (C_to_SM) when the larger bin has more particles than their baseline. We have added clarifying text to the manuscript "*Because the particle counts in the 2.5–5 μm and 5–10 μm size bins are much lower than in smaller bins, mass concentrations were used instead of number concentrations to obtain more stable and interpretable ratios.*"

11. Line 205: Parameter name is accidentally subscripted.

Response: Agreed, have made the appropriate changes.

12. Figure 1: The arrow denoting threshold2 is misleading, as it points in the opposite direction from points exceeding the threshold. If these arrow are meant to help indicate the meaning of the lines, I'd suggest removing them, and just placing the text directly adjacent to the lines.

Response: Agreed, we have made the appropriate changes.

13. Line 276: All circles are empty, not solid.

Response: Agreed, have made the appropriate changes.

14. Line 282: Does the model incorporate FEM $PM_{2.5}$, or is it only calibrated using these data?

Response: The correction does not rely on FEM PM2.5. We revised our language as follows: "*The pm2.5_alt concentration was adjusted (referred to as adj_pm2.5_alt) using a non-linear regression model in R (4.4.0) that incorporated pm2.5_alt, C_to_SM, threshold1, and a scaling factor A.*"

15. Lines 282-284: This describes that the calibration was determined using identified dust events based on the post-processing and near-real-time approaches. How would the results differ if calibration were based on identified events using the FEM-derived CF instead?

Response: We attempted to select measurements using CF. However, we have observed that even when CF>0.7, many of the pm2.5_alt estimates of PM2.5 correlate reasonably well with FEMs/FRMs (see Figure S4). In the absence of a proper dust-dominated PM definition, we decided against using a FEM-based approach for identifying measurements and for correction factor development.

16. Line 286: Parameters adj_pm2.5_alt and pm2.5_alt are accidentally subscripted.

Response: We thank the reviewer for noticing this error, and we have made the required changes.

17. Lines 301-302: Why is MBE the only metric considered? I would suggest also including a metric like Mean Absolute Error or Root Mean Square Error, including normalized variants of

these metrics, as the issue being addressed is not one of constant measurement offset, but of relative underestimation which varies in proportion to the magnitude.

Response: We have added RMSE and normalized RMSE to the analysis and discussed these metrics in Sections 3.2 and 3.3. RMSE has been added to Figures 3 and 5, Figure S14, and Table S7. Normalized RMSE is also reported in Table S7.

18. Lines 303-304: Parameters are accidentally subscripted. Furthermore, these definitions are counter-intuitive, e.g., a "positive bias" usually implies that the estimate (pm2.5_alt or adj_pm2.5_alt) is higher than the "true" value (FEM PM$_{2.5}$).

Response: We agree and have made the suggested changes. We have also changed the definition of MBE, such that it is negative when the sensor underestimates. The MBE discussion and the figures were also changed accordingly.

19. Lines 335-339: The description implies that only data identified as being associated with dust events are presented in the figure, but my understanding is that all data are shown in the figure, with black circles denoting data associated with identified dust events, via the post-processing approach. This should be clarified. Same comment applies to Lines 373-375.

Response: We agree and have made the suggested changes. The updated language now reads:

"*Figure 2 (top) compares the pm2.5_alt with FEM PM2.5 concentrations, with measurements identified as potential dust-dominated PM marked as black circles, for the 12 representative sensors.* "

"*Figure 4 (top) compares the pm2.5_alt with FEM PM2.5 concentrations, with measurements identified as potential dust-dominated PM marked as black circles, for the 12 representative sensors, using the real-time approach.* "

20. Line 403: The final statement about the impact of A is not needed; this was discussed when the correction was defined above.

Response: We agree and have removed this statement.

21. Lines 404-418: I think that the impact of using sensor-specific calibration factors discussed here is an important point to understand, and to further emphasize it, I would suggest adding a third set of points to figures 3 and 5, showing results if the sensor-specific calibration factor were used.

Response: We agree and have added the MBE and RMSE, for three different scenarios in Figure 5: A = 0.702, A = 0.998, and sensor-specific A. The results were discussed under Section 3.3.

**Reviewer 2**

1. Dust events can be a significant threat to public health, as it involves sharp increases of the coarse particles which are associated with several health hazards. The limitations of the reference air pollution measuring networks are known, as due to the high cost of the instruments used they can only be deployed in limited numbers. Low-cost sensors can provide an alternative which can potentially increase the number of monitoring points, though their usability in specific scenarios is not fully explored. This study attempts to reveal the capabilities of the low-cost sensors in measuring the coarse particles during dust events despite their inherent limitations. The study tests previously used (and new?) methodologies with measurements collected from Plantower sensors. My main objection is the title of the manuscript and the way the outcomes of the analysis are presented. The title mentions "Identification of Dust-Dominated periods", which when I first read, I was expecting that through the methodologies used dust events would be identified by the low-cost sensor data and then evaluated using the reference data. Instead, the dust events are a group of days which comply to specific rules without having any kind of evaluation (whether they were really dust events or something else, which is a possibility correctly pointed in the Introduction and Methodology parts). Similarly, in the Introduction the phrase "examines whether dust episodes can be directly identified" and in the conclusions the phrase "successfully identified" are used, which point exactly on the same thing (that the LCS provided and the regulatory instruments approved). I suggest that both the title and the parts that mention successful identification of the events are removed or at least toned down to the possible ability of the sensors to identify the events. The calibration (correction) of the sensor parts are useful though and probably should be the focus of the title and conclusions.

Response: We thank the reviewer for these thoughtful comments. Reviewer #1 expressed similar concerns. Our previous title of the manuscript was misleading, and we have changed the title to "*Correction of PM2.5 Underestimation in Low-Cost Sensors under Elevated Dust Loading Using Only Sensor Measurements*".

The objective of the manuscript was not to identify the dust-events, which would require FEM PM measurements along with meteorological parameters, but rather to identify periods with elevated dust loading, which may or may not occur during dust events. We have clarified it in the text under Section 2. We have added the following text:

"*This study focuses on identifying conditions that are specifically associated with PMS underestimation. It does not attempt to identify "dust events" in part because there is no well-defined dust event classification method based solely on FRM PM10 and PM2.5 measurements. Moreover, the PMS sensor's performance depends strongly on the underlying particle size distribution (Kaur and Kelly, 2023a; Kuula et al., 2020; Ouimette et al., 2024). Although the PMS sensors are inefficient at measuring particles with diameters > 1 μm (Kaur and Kelly, 2023a; Kuula et al., 2020; Ouimette et al., 2024), the PMS sensor can still provide reasonable estimates of PM2.5 concentrations when concentrations of particles > 1 μm in diameter are elevated, as well as particles < 1 μm in diameter.*"

In response to Reviewer 1's comment, we added a section comparing threshold-identified potential dust-dominated PM with observed dust-dominant conditions, as described below:

"*2.5.3 Evaluating the effectiveness of the two approaches in selecting dust-dominant PM conditions*

*Figure 1 and Figure S3 show that high C_to_SM tended to occur at low pm2.5_alt values, and measurements identified as potentially dust dominant using threshold1 and threshold2 generally corresponded to elevated CF values. We evaluated the effectiveness of our two approaches by examining how the potential-dust dominant conditions corresponded to various bins of FEM CF values (0–0.25, 0.25–0.5, 0.5–0.7, and >0.7). High CF values can occur under low PM10 conditions, when small PM concentrations lead to increased variability in the ratio and do not necessarily indicate dust-dominant conditions. We established criteria for "true dust-dominant PM", defined when CF > 0.7 and FEM PM10 concentrations > 100 µg/m3 and compared how PMS-derived PM2.5 (pm2.5_alt) compare with FEM PM2.5 concentrations under true dust-dominant conditions. Note that a universally accepted definition of dust-dominant conditions does not yet exist.*

*Even during periods classified as true dust events, PMS sensors can estimate PM2.5 reasonably well, as shown in Figure S4, which compares PMS-derived PM2.5 (pm2.5_alt) with FEM PM2.5 concentrations. Agreement between PMS and FEM measurements under these conditions likely reflects the presence of mixed aerosol conditions, in which coarse dust particles coexist with submicron aerosols that contribute to PM2.5 and are detectable by the PMS sensor. The primary objective of the thresholds defined here is to identify conditions under which the PMS sensor underestimates PM2.5. Accordingly, PM2.5 underestimation was defined as measurements for which pm2.5_alt was less than 0.25 times the corresponding FEM PM2.5 concentration. For the true dust-dominant periods (CF > 0.7 and PM10 > 100 µg/m3), the analysis quantified the fraction of measurements exhibiting PM2.5 underestimation that were correctly identified by the thresholds. Of the 109 sensors evaluated, 79 sensors had sufficient data to calculate CF and were included in this analysis.*"

Discussion under section 3.1:

"*Tables 1 and S5 summarize counts of true dust-dominant PM measurements and measurements with PMS PM2.5 underestimation across all sensors. For the 12 representative sensors (CF available for 8 sensors, Table 1), approximately 20% of measurements identified using our thresholds were true dust-dominant PM measurements, and roughly 16% were underestimated (pm2.5_alt / PM2.5 < 0.25). The post-processing and real-time approaches captured 68.4 ± 24.7% and 72.6 ± 22.2%, respectively, of all measurements corresponding to true dust-dominated PM measurements. When restricted to underestimated PMS PM2.5 measurements meeting the same criteria, 86.5 ± 14.9% (post-processing) and 89.7 ± 9.9% (real-time) were captured. Across the remaining 71 sensors (Table S5), 11–12% of measurements identified using our thresholds were true dust-dominant PM measurements, and 7–10% exhibited PMS PM2.5 underestimation. Although these remaining 71 sensors had a smaller fraction of true dust-dominant PM, they accounted for 60.6 ± 30% and 66.7 ± 25.6% of the underestimated PMS PM2.5 measurements, in the post-processing and real-time approaches, respectively.*"

We have also replaced the term "*dust events*" with "*dust-dominated PM*" throughout the manuscript.

We have removed the phrase "dust episodes can be directly identified" and "successfully identified" from the manuscript.

2.  Line 203. The assumed density of the particles is too low. Household dust is reported at about 1.6 and mineral dust at about 2.7. This low density would probably lead to great underestimations of the mass of the coarse particles.

Response: We agree and thank the reviewer for the comment. We only used the density during our ratio calculation (C_to_SM), which is a mass-to-mass ratios. Although we assumed a density of 1, the density cancels out in the ratio calculation. We have modified the statement in the text to clarify this "*The density was assumed to be 1 g/cm3, although density cancels out in the C_to_SM calculation.*"

In addition, we do not calculate pm2.5_alt, rather it is provided by the sensor. pm2.5_alt is widely used, and we don't want to alter how pm2.5_alt is calculated.

3.  Line 313. I don't understand the numbering scheme in reporting the number of potential dust events. What do 0 – 3785 or 0 – 9% mean? Does it start from zero because they are "potential"?

Response: We have changed the wording for clarification. The numbers 0 – 3785 are the counts of hourly measurements that get selected as potential dust-dominated PM measurements, and the 0 – 9% are the % of hourly measurements that get selected as potential dust-dominated PM (out of each sensor's total hourly measurements).

Minor comments

4.  Please add the six size bins reported by the PMS sensors in chapter 2.1 when the sensors are presented.

Response: We agree and have added these bins. The manuscript now reads: "*The sensor converts the total light scattering into several different metrics, including particle counts in six bins (>0.3 um, >0.5 um, >1 um, >2.5 um, >5 um, and >10 um),* "

5.  In line 146 it is mentioned that the data is downloaded by the company's site. Are these given as raw or calibrated data?

Response: The reviewer makes a good point. The data is raw data. We have added the word raw to this statement.

6.  In line 161 the variable pm2.5_cf_1 is mentioned. What is it (according to the manufacturer)? How is it named and presented in the datasets?

Response: The pm2.5_cf_1 refers to one of the pm2.5 measurements reported by the sensor, which uses a correction factor of one (CF= 1). We have added text clarifying "*pm2.5_cf_1 (PM2.5 mass concentration reported by PMS sensor using a correction factor = 1)*". We did not use the pm2.5_cf_1 in the analysis; we only used pm2.5_alt.

7. While out of the scope of the specific paper, but were there any discrepancies from the different FEM measurements reported at the same sites (as mentioned in line 181)?

Response: We certainly did see differences between different FEMs in this study. Other studies (Le et. al, https://doi.org/10.1016/j.atmosenv.2019.117138; Khan et. al., https://doi.org/10.3390/atmos15080978; Chadwick et. al. https://doi.org/10.1016/j.jaerosci.2021.105766) have also observed this discrepancy. We agree that this is beyond the scope of this manuscript. We did clarify how we selected which FEM measurements to use when multiple FEMs were available (See Section 2.3).

8. Line 205. $CF_{toUF}$ is inconsistent with the naming used in other spots (CF_to_UF).

Response: We thank the reviewer for noticing this error, and we have made the required changes. The CF_to_UF has been changed to C_to_SM to reflect the ratio of coarser to submicron fraction.

9. How did the regression perform in the correction at chapter 2.5.3?

Response: We thank the reviewer for the comment. We have reported the R2 value in the text and also as a table in the supplementary material. The following text has been added to section 3.2 and 3.3.

Section 3.2 "*The coefficient of determination ($R^2$), increased from 0.290 for pm2.5_alt versus FEM PM2.5 to 0.653 for adj_pm2.5_alt versus FEM PM2.5. Sensor-specific changes in $R^2$ are presented in Table S6.*"

Section 3.3 "*The coefficient of determination ($R^2$), irrespective of the A values, increased from 0.262 for pm2.5_alt versus FEM PM2.5 to 0.513 for adj_pm2.5_alt versus FEM PM2.5. Sensor-specific changes in $R^2$ are presented in Table S6.*"

10. In line 365, please use the full names of the states.

Response: We agree and have added the full names.

11. In line 442, please use Celsius as well (can be put in parentheses).

Response: We agree and have added the Celsius.

12. Line 451, while I like the optimism, a major shortcoming with the use of low-cost sensors is the variability of their performance in time. A factory "Factor A" would only be true for a

specific period (as pointed by figure S5), though it would be a proxy of its general performance. In general, though it would be an improvement from the "tabula rasa" that every sensor comes as. Consider adding a small asterisk about this.

Response: We agree that sensor performance can vary over time. We agree that this would be a limitation of a laboratory-based approach, and we adding the following clarifying statement: "*It could significantly enhance the accuracy of real-time dust-dominated PM measurement detection and correction although a strategy for addressing sensor performances changes over time would still be needed.*"